# Indirect Competitive ELISA for the Determination of Total Chromium Content in Food, Feed and Environmental Samples

**DOI:** 10.3390/molecules27051585

**Published:** 2022-02-27

**Authors:** Xiaofei Wang, Yanan Wang, Shuyun Wang, Jie Hou, Linlin Cai, Guoying Fan

**Affiliations:** 1Xinxiang Institute of Engineering, College of Bioengineering, Xinxiang 453700, China; wangxiaofei1119@126.com (X.W.); wangshuyun5166@163.com (S.W.); 2Henan Institute of Science and Technology, College of Animal Science and Veterinary Medicine, Xinxiang 453003, China; wyn564@126.com (Y.W.); nickhou0199@163.com (J.H.); cailinlin0503@163.com (L.C.)

**Keywords:** trivalent chromium ions, immunogen, monoclonal antibody, indirect competitive ELISA, total chromium mass concentration

## Abstract

**Background**: This study aimed to prepare monoclonal antibodies (mAbs) with high immunoreactivity, sensitivity, and specificity for the chelate (Cr(III)-EDTA) of trivalent chromium ion (Cr(III)) and ethylenediamine tetraacetic acid (EDTA). Further, the study established an indirect competitive enzyme-linked immunosorbent assay (icELISA) for detecting the total chromium content in food, feed, and environmental samples. **Methods**: Hapten Cr(III)-iEDTA was synthesized by chelating Cr(III) with isothiocyanatebenzyl-EDTA (iEDTA). Immunogen Cr(III)-iEDTA-BSA formed by chelating Cr(III)-iEDTA with bovine serum albumin (BSA), and coating antigen Cr(III)-iEDTA-OVA formed by chelating Cr(III)-iEDTA with ovalbumin (OVA) were prepared using the isothiocyanate method and identified by ultraviolet spectra (UV) and inductively coupled plasma optical emission spectrometry (ICP-OES). Balb/c mice were immunized with the Cr(III)-iEDTA-BSA, and the anti Cr(III)-EDTA mAb cell lines were screened by cell fusion. The Cr(III)-EDTA mAbs were prepared by induced ascites in vivo, and their immunological characteristics were assessed. **Results**: The immunogen Cr(III)-iEDTA-BSA was successfully synthesized, and the molecular binding ratio of Cr(III) to BSA was 15.48:1. Three hybridoma cell lines 2A3, 2A11, and 3D9 were screened, among which 2A3 was the best cell line. The 2A3 secreted antibody was stable after six passages, the affinity constant (Ka) was 2.69 × 10^9^ L/mol, its 50% inhibition concentration (IC50) of Cr(III)-EDTA was 8.64 μg/L, and it had no cross-reactivity (CR%) with other heavy metal ion chelates except for a slight CR with Fe(III)-EDTA (1.12%). An icELISA detection method for Cr(III)-EDTA was established, with a limit of detection (LOD) of 1.0 μg/L and a working range of 1.13 to 66.30 μg/L. The average spiked recovery intra-assay rates were 90% to 109.5%, while the average recovery inter-assay rates were 90.4% to 97.2%. The intra-and inter-assay coefficient of variations (CVs) were 11.5% to 12.6% and 11.1% to 12.7%, respectively. The preliminary application of the icELISA and the comparison with ICP-OES showed that the coincidence rate of the two methods was 100%, and the correlation coefficient was 0.987. **Conclusions**: The study successfully established an icELISA method that meets the requirements for detecting the Cr(III)-EDTA chelate content in food, feed, and environmental samples, based on Cr(III)-EDTA mAb, and carried out its preliminary practical application.

## 1. Introduction

Heavy metal chromium (Cr) is an important modern industrial strategic resource widely used in metallurgy, fire resistance, chemical industry, national defense, and other fields. The metal has become a serious pollutant in the environment and food due to excessive use and a lack of corresponding protection measures [1]. Based on the analysis of 1625 sites and 1799 literatures, Li et al. [2] reported a nationwide assessment of Cr pollution in agricultural soils in China for the first time. The Cr concentration in farmland soil ranged from 1.48 to 820.24 mg/kg, and the Cr concentration at about 4.31% and 0.12% of the sampling points exceeded the screening value (150 mg/kg) and the control value (800 mg/kg) (GB 15618-2018), respectively. In September 2014, the “chromium poison capsule” incident occurred in Zhejiang Province, China. The Cr levels exceeded the standard by 65 times, which once again aroused the widespread concern of the government and society about excessive Cr pollution. Under natural conditions, Cr exists in a variety of oxidation states, but the most common and stable oxidation states are trivalent chromium (Cr(III)) and hexavalent chromium (Cr(VI)). Cr(III) often exists as the less toxic chromite (FeOCr_2_O_3_) while Cr(VI) is often present as the more toxic chromate (CrO_4_^2−^) or dichromate (Cr_2_O_7_^2−^), which is 100 times more toxic than Cr(III) to organisms [3]. Under normal physiological conditions, Cr(VI) enters cells after ingestion and can be reduced to Cr(V), Cr(IV), and Cr(III), which can change the activity of free radicals such as sulfur groups and hydroxyl in the body, and which destroys the integrity of the cells by attacking proteins, DNA and membrane lipids [4]. Since Cr(VI) has hepatotoxicity [5], nephrotoxicity [6], genotoxicity [7], neurotoxicity [8] and carcinogenic toxicity [9] to human health, it is listed as one of the eight chemical substances with the greatest harm to the human body and is one of the three internationally recognized carcinogenic metal poisons [10]. In order to strictly control Cr pollution in food, the United States Environmental Protection Agency (EPA), the World Health Organization (WHO) and the European Community (EC) set the maximum limit of Cr in drinking water to 100 μg/L (EPA 822-R-06-013), 50 μg/L (WHO ISBN 9241546743) and 50 μg/L (98/83/EC), respectively [11]. The Chinese national standard (GB 2762-2017 National Food Safety Standard-Limits of Contaminants in Food) stipulates that the maximum Cr limit should be less than 2.0 mg/kg, 1.0 mg/kg and 0.5 mg/kg for aquatic animals and their products, grains, beans, meat and their respective products, and vegetables and their products, respectively [12].

Currently, a number of methods are mainly used for the detection of Cr ions, including graphite furnace atomic absorption spectrometry (GFAAS) [13], flame atomic absorption spectrometry (FAAS) [14], inductively coupled plasma mass spectrometry (ICP-MS) [15] and inductively coupled plasma optical emission spectrometry (ICP-OES) [16]. Although these methods have high precision, they require expensive instruments, have high technical requirements and testing costs, have complicated sample pre-processing, and therefore, cannot be used to perform on-site testing, and this limits their application [17]. The indirect competitive enzyme-linked immunosorbent assay (icELISA) based on specific and sensitive antigen-antibody reaction is a mature and advanced detection method developed in recent years. Its principle is briefly described as follows. Because the microplate has protein adsorption ability, the antigen can be coated on the microplate to form a solidified antigen. The solidified antigen, the hapten to be tested, and the specific antibody are incubated together. Since the solidified antigen and the hapten to be tested have a single and the same antigen epitope, they can compete for the antigen epitope on the specific antibody. After washing many times, the formed solidified antigen-antibody complex is retained on the microplate, while the hapten and antibody complex is washed off. The retained solidified antigen-antibody complex reacts with the enzyme labeled secondary antibody to form a new antigen-antibody complex with enzyme activity, and then the enzyme reaction substrate is added. Because the solidified immune complex is inversely proportional to the concentration of the small molecule hapten in the sample, the content of the small molecule hapten to be tested can be determined according to the color development or absorption value [18,19]. Compared with physical and chemical analysis methods, such as GFAAS, FAAS, ICP-MS, ICP-OES, etc., and other immunoassays such as gold immunochromatographic assay (GICA), fluorescence polarization immunoassay (FPIA), etc., although the icELISA has some defects—such as it is easily affected by environmental and reaction conditions, poor stability, strong nonspecific reaction, prone to false positive results, and strict requirements for sample pretreatment—it has become a fundamental method in the detection of heavy metal ion chelates due to its strong selectivity, high sensitivity, speed and simplicity, large sample screening volume and field operation. Up to now, icELISA methods for the determination of Cd(II) [20,21], Hg(II) [22,23], Pb(II) [24] and so on have been established. Sasaki et al. [25] developed a Cr ion immunodetector with a limit of detection (LOD) of 1.6 μg/L; Liu et al. [11]and Zou et al. [26] developed a Cr ion colloidal gold immunochromatographic test strip with a LOD of 50 μg/L and 0.1 μg/L, respectively; and Yao et al. [17] developed a Cr(III) plasma ELISA with a LOD of 3.13 μg/L. However, the development of an ELISA kit for Cr ion detection has not yet been reported. The study prepared a Cr(III) monoclonal antibody (mAb) and ethylenediamine tetraacetic acid (EDTA) chelate (Cr(III)-EDTA) with high specificity and high sensitivity, and developed an indirect competitive ELISA (icELISA) kit for the detection of total Cr ion contamination residues in food, feed, and environmental samples.

## 2. Results

### 2.1. UV Identification

In the UV range between 220–400 nm, BSA has a characteristic peak at 278 nm, Cr(III)-iEDTA, which is conjugate of Cr(III) and isothiocyanatebenzyl-EDTA (iEDTA), has two characteristic peaks at 264 and 332 nm, and Cr(III)-iEDTA-BSA has two characteristic peaks at 272 and 332 nm, which contains, simultaneously, the characteristic peaks of BSA and Cr(III)-iEDTA (Figure 1). The results indicated that the immunogen synthesis was successful and based on the Lambert–Beer law, the calculated molar ratio of Cr(III)-iEDTA to BSA was 15.45:1.

### 2.2. Determining the BSA and Cr(III) Content

The BSA content in Cr(III)-iEDTA-BSA measured by DU-800 ultraviolet-visible spectrophotometer was 5.81 mg/mL. The Cr(III) content in Cr(III)-iEDTA-BSA measured using an Optima 2100 DV ICP-OES was 70.4 μg/mL. The results showed that the immunogen synthesis was successful, and the calculated molar ratio of Cr(III) to BSA was 15.48:1, consistent with the UV identification results.

### 2.3. Selection for Potential Cell Fusion in Immunized Mice

The titers and the 50% inhibition concentration (IC50) values of Cr(III)-EDTA polyclonal antibodies (pAbs) derived from the five immunized mice after the five inoculations were determined by both an indirect non-competitive ELISA (inELISA) and an indirect competitive ELISA (icELISA). The five immunized mice had a strong positive immune response, and the titers exceeded 1: (1.6 × 10^3^), in which the immune response of mouse No.4 was the most effective where the titers reached 1: (5.12 × 10^4^) (Figure 2). Meanwhile, all five immunized mice could recognize Cr(III)-EDTA well, and among them, the IC50 value of mouse No.4 was the lowest (24.8 μg/L) (Figure 3). Owing to the highest titers and lowest IC50 value, mouse No.4 was selected for the next cell fusion step.

### 2.4. Screening of Positive Clones and Establishment of Hybridoma Cell Lines

Twelve days following the cell fusion procedure, growing hybridoma cell clones could be observed by the naked eye. There were 338 wells formed by hybridoma cell clones in 384 wells of the four cell culture plates, and the fusion rate of the mouse spleen cells with myeloma cells was approximately 88%. The inELISA was employed to determine their positive reaction and 56 wells of the 338 wells with clone formation had a positive reaction with a positive rate of 16.3%. The icELISA was used to determine their recognition ability against Cr(III)-EDTA and 8 wells of the 56 positive wells had strong recognition ability with a strong positive rate of 14.3%. After subcloning thrice using a limiting dilution, three hybridoma cell lines with strong immune response and recognition ability were screened and named 2A3, 2A11 and 3D9, which were used to produce mAbs. The titers and IC50 values of all three mAbs in culture supernatants and ascites were determined by inELISA and icELISA (Table 1).

### 2.5. Analysis of Isotype and Stability of Hybridoma Cell Lines

A mouse mAb isotyping kit was adopted to determine that 2A3 and 2A11 mAbs belonged to the IgG1 isotype with a kappa light chain, and 3D9 mAb was of the IgG1 isotype and possessed a lambda light chain (Figure 4). The absorption values of the cell culture supernatants before cryopreservation of three hybridomas were basically consistent with that after five times of cryopreservation, resuscitation, and subculture, indicating that three hybridomas could stably secrete antibodies (Figure 5).

### 2.6. Determining the Affinity of Three Cr(III)-EDTA mAbs

The Ka of Cr(III)-EDTA mAb 2A3, 2A11, and 3D9 were 2.69 × 10^9^, 1.03 × 10^9^ and 7.85 × 10^8^ L/moL, respectively (Figure 6). Since 2A3 had the highest affinity, it was selected for further experimentation to develop icELISA.

### 2.7. IcELISA Optimization and Establishment of IcELISA Standard Curve

#### 2.7.1. IcELISA Optimization

The chessboard titration results indicated that the optimal concentration of the coating antigen Cr(III)-iEDTA-OVA was 2.0 µg/mL, mAb 2A3 was 0.6 µg/mL (1:20,000), and GaMIgG-HRP was 0.6 µg/mL (1:1000) (data not shown). Four combinations of mAb dilution buffer and standard preparation buffer were performed to investigate the effect of the assay buffer. As shown in Table 2, when Cr(III)-EDTA and 2B6 mAb were dissolved in Hepes buffer solution (HBS), IC50 was slightly decreased and Amax/IC50 was slightly increased, indicating that the K^+^ concentration had a certain effect on icELISA performance. Therefore, HBS was chosen as assay buffer. The effect of HBS buffer ionic strength on icELISA is shown in Figure 7. The results showed that with the continuous increase in ionic strength, the Amax/IC50 gradually decreased while the IC50 value gradually increased, i.e., the sensitivity of icELISA decreased. Therefore, the best minimum NaCl content in HBS buffer was 137 mM; hence, there was no need to increase the ionic strength. The effects of the pH values on the icELISA are shown in Figure 8. The pH values between 5.0 and 9.0 had a significant effect on the Amax/IC50 and IC50 values. However, the Amax/IC50 was highest and the IC50 value was lowest at a pH of 7.4, indicating full binding between the antibodies and the antigen. Therefore, the HBS with a pH value of 7.4 was selected for an icELISA.

#### 2.7.2. IcELISA Standard Curve

As per the above-optimized conditions, the standard curve of Cr(III)-EDTA icELISA was established (Figure 9). The derived linear regression equation was y = −33.906x + 18.239 and IC50 value was 8.64 μg/L.

### 2.8. Validation of IcELISA

#### 2.8.1. Sensitivity

Based on the results of six repeated determinations of 20 different blank samples, the theoretical LOD of the icELISA was 0.92 μg/L, but considering the actual detection needs and operational errors, the LOD was determined as 1.0 μg/L. The working range (IC20 to IC80) of the icELISA was 1.13 to 66.30 μg/L calculated by the above regression equation.

#### 2.8.2. Specificity

The CRs of the icELISA were determined with other metal ions, including Cd(II)-EDTA, Hg(II)-EDTA, Pb(II)-EDTA, Cu(II)-EDTA, Mn(II)-EDTA, Mg(II)-EDTA, Zn(II)-EDTA, Fe(III)-EDTA, and Al(III)-EDTA. Concentrations of the different metal ions ranging from 0 to 10^4^ μg/L were prepared and converted to metal–chelate complexes with chelating of EDTA for the assessment by the optimized assay (Table 3). The results showed that the CR values of the icELISA with other metal ions were below 0.5% except for a slight reactivity with Fe(III)-EDTA. 

#### 2.8.3. Accuracy and Precision

The accuracy and precision results obtained from each spiked sample measured by the optimized assay are shown in Table 4. The statistical results showed that the average intra-assay recoveries of soil, wheat flour, rice, and pig feed samples were 109.5%, 94.8%, 94.5% and 90%, respectively, while the average inter-assay recoveries of the samples were 94.3%, 95.2%, 97.2% and 90.4%, respectively, indicating that the icELISA had high accuracy. The average intra-assay CVs of the samples were 12.5%, 11.5%, 11.6% and 12.6%, respectively, while the inter-assay CVs were 12.6%, 11.3%, 11.1% and 12.7%, respectively, which all were less than 15%, indicating that icELISA had high precision.

#### 2.8.4. Reliability

The results of 80 actual samples detected by icELISA and ICP-OES were compared as shown in Table 5. Out of 80 samples, 13 positive samples were detected with a positive rate of 16.3%, consistent with the detection results of ICP-OES, and had a coincidence rate of 100%. The icELISA positive value of 13 positive samples was 1.2 to 44.2 μg/L, the ICP-OES positive value was 1.1 to 42.3 μg/L, the CVs was 10.6% to 13.3%, and the coefficient correlation of results between icELISA and ICP-OES was 0.987, which indicated that the developed icELISA was reliable for the detection of Cr(III) in actual samples.

## 3. Discussion

### 3.1. The Synthesis Method of Cr(III) Immunogen

High-quality immunogens form the basis for preparing high-quality antibodies, and they must have three characteristics, i.e., high immunogenicity, full exposure of antigen epitopes, and non-toxicity to immunized animals. However, Cr(III) is too small in size (ionic radius 0.615 Å), and its structure is too simple to directly induce a specific immune response in the body. Furthermore, the charged charge can irreversibly react with the biological molecules in the body and cause poisoning. Therefore, immunogen preparation was a critical step in this study. Guo et al. [27] and Blake et al. [28] reported that Cr(III) must be coupled with a carrier protein to synthesize immunogens through the chelation of a chelating agent. As an ideal chelating agent of metal ion immunogens, EDTA could selectively chelate metal ions and weaken the reaction ability between metal ions and biomolecules, contained active groups that react with the carrier protein, and the conjugate could be recognized by the immune system. Through a large number of screening and comparative experiments, macromolecular bifunctional chelating agents were the most ideal coupling agents. At present, the two types of commonly used macromolecular bifunctional chelating agents include EDTA and its derivatives, such as nitrophenyl-EDTA, cyclohexyl-EDTA, phenyl isothiocyanate-EDTA, etc., and diethylenetriamine pentaacetic acid (DTPA) and its derivatives, such as nitrophenyl-DTPA, cyclohexyl-DTPA, phenyl isothiocyanate-DTPA, etc. Zhu et al. [29] and Xiang et al. [24] synthesized Cd(II) and Pb(II) immunogens with DTPA as a coupling agent, respectively, and achieved a good immune effect. However, Liu et al. [30] and Leopold et al. [31] used iEDTA as a coupling agent to synthesize Cu(II) and Hg(II) immunogens, and also achieved a good immune effect. It was considered that iEDTA could form a stable six dentate coordination compound with metal ions due to its more complex structure and higher activity compared with DTPA. The antigen epitopes were more fully exposed and easier to be recognized by B lymphocytes to produce specific antibodies. Therefore, in this experiment, iEDTA was selected as the preferred coupling agent, and the Cr(III) immunogen was prepared by the isothiocyanate method.

### 3.2. The Sensitivity of IcELISA for Cr(III)-EDTA

A high-quality antibody is the core reagent for establishing immunoassays. The quality of antibodies is mainly reflected in the sensitivity and specificity of the antigen–antibody reaction mode. The sensitivity is determined by the affinity of the reaction between the antibody and its corresponding antigen, which reflects the binding strength between the antibody and the antigen and is expressed by the affinity constant (Ka). The Ka of Cr(III)-EDTA mAb 2A3 with the highest affinity was 2.69 × 10^9^ L/mol as determined by the Batty saturation method. Velanki et al. [32] showed that a Ka of 10^7^ to 10^12^ L/mol was a high affinity antibody, whereas a Ka of 10^5^ to 10^7^ L/mol was a low affinity antibody, and therefore, the Cr(III)-EDTA mAb 2A3 developed in this study was a high affinity antibody. The Cr(III)-EDTA icELISA detection method was established based on mAb 2A3. To further improve the sensitivity of the assay, three main parameters of the icELISA were optimized. First, the checkerboard method was employed to determine the optimal working concentrations of the coating antigens, and Cr(III)-iEDTA-OVA, mAb 2A3 and GaMIgG-HRP were 2.0 µg/mL, 0.6 µg/mL, and 0.6 µg/mL, respectively. Additionally, the appropriate ionic strength of the buffer was optimized, and the minimum NaCl content in the HEPES-KCl buffer was 137 mM. Lastly, the appropriate HBS pH value was optimized when the pH was 7.4; the Amax/IC50 value was the highest and the IC50 value was the lowest, indicating that the antibody binds to the antigen sufficiently, and the optimal HBS pH value of the icELISA system was 7.4.

### 3.3. The Specificity of IcELISA for Cr(III)-EDTA

Specificity reflects the ability of an antibody to recognize an antigen or a hapten, which is determined by the complementarity between the spatial structure of the antibody hypervariable region and the antigen determinant, expressed by a cross-reactivity value. Alzari et al. [33] confirmed that the range of antigen conformational epitopes recognized by antibodies was 160 to 900 Å; the ionic radius of Cr(III) was 0.615 Å and was not recognized by antibodies. The Cr(III)-EDTA belonged to the hapten and could specifically bind with antibodies. Delehanty et al. [34] further revealed the molecular mechanism of metal ion chelate hapten antibody recognition and identified three antibody residues through homology modeling and mutation. This included Trp52 and His96 in the heavy chain and Arg96 in the light chain that played an important role in the specific recognition of metal ion chelate. Among them, His96 mediated the direct recognition of metal ion ions, Trp52 participated in the hydrophobic accumulation with the benzyl part of the chelating agent, and Arg96 mediated the electrostatic and hydrogen bonds of the chelate part. The antibody recognition of the metal chelating hapten occurred through a limited number of molecular contacts, and these molecular interactions involved the direct connection between the antibody and metal ions and the interaction between the antibody and chelating agent. The Cr(III)-EDTA mAb 2A3 prepared in this study had no CR with other metal ion chelates except for the slight CR (1.21%) with Fe(III)-EDTA. The ionic radii of Cr(III) and Fe(III) are 0.615 Å and 0.645 Å, respectively, with a difference of 0.3 Å. The stereoisomeric conformation and the difference distance matrixes of Cr(III)-EDTA and Fe(III)-EDTA are very close, hence Cr(III)-EDTA mAb 2A3 can slightly recognize the Fe(III)-EDTA [11].

### 3.4. The Selection of ELISA Format

Currently, there is no unified standard for the classification of ELISA in academic circles, and different literatures have different classification opinions on the principle or operation, but most scholars prefer to be divided into inELISA, direct non-competitive ELISA (dnELISA), icELISA, direct competitive ELISA (dcELISA) and sandwich ELISA. Different types of ELISA have different functions, and among them, the function of icELISA can be used to detect antibodies, antigens and haptens. Since a small molecule hapten has only one epitope and needs to be detected in competitive format, sandwich ELISA requiring two or more epitopes, and inELISA and dnELISA in non-competitive format cannot be used; therefore, icELISA and dcELISA are mainly used [35,36].

The ELISA models for the detection of small molecule hapten compounds mainly include icELISA and dcELISA. According to the reaction principle of ELISA, they are the same in three aspects: first, they are based on the specific antigen–antibody reaction; second, they all use the known amount of limited antibody, which forms a competitive relationship between different antigens and antibodies; third, they all use the enzymatic system, and the detection results are negatively correlated with the content of the hapten to be tested [36,37]. Their differences lie in three aspects: first, the reaction formats are different. The icELISA is that the solidified antigen and free hapten compete for the epitope of a specific antibody, and the adsorbed solidified antigen-antibody complex react with the enzyme-labeled secondary antibody to develop color; the dcELISA is that free hapten and the enzyme-labeled hapten compete for the epitope of the immobilized specific antibody, and the adsorbed solidified antibody and enzyme-labeled hapten complex are colored by the enzymatic system. Second, the solidified reagent and enzyme-labeled reagent are different. The icELISA uses the solidified self-made coating antigen and the standardized enzyme-labeled secondary antibody, and the dcELISA uses the solidified self-made specific antibody and the self-made enzyme-labeled hapten. Third, the detection performances are different. The icELISA is more sensitive than that of the dcELISA for two reasons. One is that in icELISA, the contact area between the solidified antigen and antibody is small, the probability of a reaction with the antibody is small, and the probability of a reaction between the solid-phase antigen and hapten to be tested and antibody is unequal, showing that the slope of inhibition curve is large, and the sensitivity is high. On the contrary, in dcELISA, both the enzyme-labeled hapten and the free hapten to be tested are in liquid phase, and the probability of a reaction with the antibody is equal, its slope of inhibition curve is small, and the sensitivity is low. The second point is that in the system amplification, the enzyme-labeled secondary antibody is used in icELISA, since the enzyme-labeled secondary antibody can target multiple parts of the antibody, there is an amplification effect, which can improve the sensitivity, while there is no amplification effect in dcELISA format. However, dcELISA has the advantages of a short reaction time and no need to use a commercial enzyme-labeled secondary antibody [38,39]. In our previous experiments, we also conducted similar comparative studies [40]. In contrast, because the enzyme-labeled secondary antibody in icELISA format is a standardized reagent, the establishment of icELISA is relatively simple, stable, and sensitive, while the enzyme-labeled hapten in dcELISA mode is a self-made reagent, the cross-linking methods of hapten to enzyme are different, and the activity and stability of the enzyme are also different, so the establishment of the dcELISA method is more difficult and lacks stability. However, icELISA also has defects such as cumbersome operation steps, a long detection time, and high detection cost. Although it can be used to detect a large number of samples, it is more suitable for professional laboratories and testing institutions.

## 4. Materials and Methods

### 4.1. Chemicals and Reagents

The standard metal ion solution in salt form of Cr(III), Cd(II), Hg(II), Pb(II), Cu(II), Mn(II), Mg(II), Zn(II), Fe(III), and Al(III) were purchased from the China National Standard Material Center (Beijing, China). 1-(4-isothiocyanobenzyl) ethylenediamine-*N*,*N*,*N*′,*N*′-tetraacetic acid (iEDTA) was obtained from Dojindo Laboratories (Kumamoto, Japan). Ethylenediamine tetraacetic acid (EDTA), polyethylene glycol 1500 (PEG 1500, 50%), hypoxanthine aminopurine thymidine (HAT), hypoxanthine thymidine (HT), 4-hydroxyethyl piperazine ethanesulfonic acid (HEPES), phenacetin, 3,3,5,5-tetramethylbenzidine (TMB), urea peroxide and Tween-20 were purchased from Sigma Aldrich (St. Louis, MO, USA). Bovine serum albumin (BSA, molecular weight (MW) 66430), ovalbumin (OVA, MW43000), Freund’s complete adjuvant (FCA), Freund’s incomplete adjuvant (FIA), and L-glutamine containing medium RPMI-1640 were obtained from Pierce (Rockford, IL, USA). Horseradish peroxidase-conjugated goat anti-mouse IgG (GaMIgG-HRP) and mouse monoclonal antibody isotype kits were purchased from Sino-American Biotechnology Company (Shanghai, China). Fetal Bovine Serum (FBS) was produced by Hangzhou Sijiqing Bioengineering Materials Co., Ltd. (Hangzhou, China). All other solvents, reagents and chemicals were standard commercial products above analytical grade. The cell culture plates (6 wells, 24 wells and 96 wells) and culture flasks were purchased from Costar (Bethesda, Rockville, MD, USA). The 96-well transparent polystyrene microplates were obtained from the Jiangsu Boyang Experimental Equipment Factory (Nanjing, China).

### 4.2. Buffer and Hybridoma Culture Medium

The buffers used for ELISA were as follows: (1) phosphate buffer solution (0.01 M PBS, pH 7.4) composed of NaCl (137 mM), Na_2_HPO_4_·12H_2_O (10 mM), KCl (2.68 mM) and KH_2_PO_4_ (1.47 mM). (2) The carbonate buffer solution (0.05 M CBS, pH 9.6) was composed of Na_2_CO_3_ (15 mM) and NaHCO_3_ (35 mM), which was mainly used to coat antigen or the antibody on microplates. (3) The washing buffer was PBS containing 0.05% Tween-20 (PBST). (4) The blocking buffer was PBS containing swine serum 5% (*v*/*v*). (5) The substrate buffer was a mixture of part A (500 mL) and part B (500 mL) solutions. Part A contained 3.15 g citric acid, 6.966 g anhydrous sodium acetate, 0.08 g phenacetin, and 0.05 g urea peroxide adjusted to a pH of 5.0 using HCl per 1 L of water, while Part B contained 1.27 g of TMB dissolved in 500 mL of methanol and 500 mL of glycerol. (6) The stop solution was 2 M H_2_SO_4_.

The hybridoma culture media used for cell culture were as follows: (1) 78 mL of RPMI-1640 medium, 20 mL of FBS, 1.0 mL of antibiotics, and 1.0 mL of Hepes buffer solution (HBS, 0.01 M, pH7.4, containing 10 mM HEPES, 137 mM NaCl, and 3 mM KCl). (2) The cell freezing solution was composed of dimethyl sulfoxide (DMSO, 10%, *v*/*v*) in a complete medium.

### 4.3. Experimental Animals and Cells

The 6-week-old female Balb/c mice were provided by the Experimental Animal Center of Medical College of Zhengzhou University with the animal license number of scxk (Yu) 2010-0002. The animals were housed at 24 ± 2 °C, and a relative humidity of 50 ± 20%, with a 12 h light and dark cycle constantly maintained throughout the experiment. The experimental animals were randomly given tap water and fed ad libitum. NS0 myeloma cells were obtained from the Key Laboratory of Animal Immunology of the Ministry of Agriculture (Zhengzhou, China).

### 4.4. Equipment and Instruments

A Multiskan MK3 microplate reader (Shanghai Thermal Power Company, Shanghai, China) was used to measure the absorbance of the microplates. The UV spectrum of the immunogen was measured by a DU-800 ultraviolet-visible spectrophotometer (Beckman-Coulter, Fullerton, CA, USA). A Galaxy S CO_2_ incubator (RS-Biotech, Ayrshire, UK) was used for cell culture and a TS100-F inverted microscope (Nikon, Tokyo, Japan) was used for cell observation. An Optima 2100DV inductively coupled plasma optical emission spectrometer (ICP-OES) (PerkinElmer Inc., Waltham, MA, USA) was used to determine the Cr(III) content in the immunogenic Cr(III)-iEDTA-BSA and verification test. Exceed DZG-303A ultrapure water system was purchased from Chengdu Corning Special Experimental Pure Water Equipment Factory (Chengdu, China), an LDZX-30KB vertical pressure steam sterilizer was purchased from Shanghai Shen’an Medical Instrument Factory (Shanghai, China), and an SW-CJ-2FD ultra clean workbench was purchased from Suzhou Purification Equipment Co., Ltd. (Suzhou, China).

### 4.5. Immunogen Synthesis

The immunogen Cr(III)-iEDTA-BSA was synthesized following a modified isothiocyanate method described by Ling et al. [20] and Zhou et al. [41]. A total of 10 mg of iEDTA was dissolved in 1.0 mL of dimethyl sulfoxide (DMSO) to form a metal chelating agent solution, and 18.1 mg of CrCl_3_·6H_2_O was dissolved in 1.0 mL of HBS to form a Cr(III) solution. After mixing the metal chelating agent solution and the Cr(III) solution, the pH value of the resulting solution was adjusted with NaOH to 7.0 and left for 12 h to form a Cr(III)-iEDTA chelate hapten. A total of 20 mg BSA was dissolved in 1.0 mL of HBS to form a 20 mg/mL BSA solution. Then, 1 mL Cr(III)-iEDTA chelate hapten solution was added to 1.0 mL BSA solution and the pH value was adjusted to 9.0 with NaOH. The reaction solution was stirred at room temperature for 24 h and dialyzed against PBS at 4 °C for 10 d. The dialysate was changed daily, collected, and stored at −20 °C for later use. The synthetic route of Cr(III)-iEDTA-BSA is shown in Figure 10. The coating antigen Cr(III)-iEDTA-OVA was prepared via the same method.

### 4.6. Immunogen Identification

#### 4.6.1. UV Identification

The HBS was used to configure a 1.0 mg/mL BSA solution and a Cr(III)-iEDTA-BSA solution with a 1.0 mg/mL BSA concentration, and the UV scanning was performed in the wavelength range of 220–400 nm. The molecular binding ratio of Cr(III)-iEDTA to BSA was calculated based on the Lambert–Beer law. The calculation equation was as follows:A = εCL.(1)
where A is the absorbance value, ε is the molar extinction coefficient, (constant value), C is the solute concentration, and L is the optical path.

#### 4.6.2. Determining the Mass Concentration of BSA and Cr(III)

The Cr(III)-iEDTA-BSA was diluted with PBS times ratio, and the mass concentration of BSA in Cr(III)-iEDTA-BSA was measured by the DU-800 ultraviolet-visible spectrophotometer at 278 nm wavelength. The Cr(III)-iEDTA-BSA was diluted 50 times with PBS, the mass concentration of Cr(III) in Cr(III)-iEDTA-BSA measured by an Optima 2100DV ICP-OES, and the wavelength of 267.716 nm was adapted as the analysis line of Cr(III) [24]. The molecular binding ratio of Cr(III)-iEDTA to BSA was calculated using the equation [42]:The molecular binding ratio = M_hm_/M_pro_.(2)
where M_hm_ is the molar concentration of heavy metal and M_pro_ is the molar concentration of BSA.

### 4.7. Preparation of Cr(III)-EDTA mAb

#### 4.7.1. Potential Mouse Selection for Cell Fusion

Five-week-old female Balb/c mice were immunized with immunogen Cr(III)-iEDTA-BSA. The immunization method was subcutaneously injected into the neck at multiple sites, the immunization dose was 100 μg/head, calculated according to the amount of BSA in Cr(III)-iEDTA-BSA, and the mice were immunized once every four weeks, five times. For the first immunization, Cr(III)-iEDTA-BSA was dissolved with sterilized PBS and mixed with the same amount of FCA for emulsification. For enhanced immunization, Cr(III)-iEDTA-BSA was dissolved with sterilized PBS and mixed with the same amount of FIA for full emulsification. The tails were cut off 21 days after the last immunization to collect blood, and the serum was separated to obtain the Cr(III)-EDTA polyclonal antibody (pAb). The titers of Cr(III)-EDTA pAb; which represent immune reactivity, were determined by an indirect non-competitive ELISA (inELISA), and the IC50 value was determined by an icELISA. The mouse with the highest titers and the lowest IC50 value was selected four days before cell fusion and administered with an intraperitoneal booster injection of 200 µg of Cr(III)-iEDTA-BSA without any adjuvant. The mouse was then sacrificed, and the spleen was harvested to obtain the hybridomas.

#### 4.7.2. Cell Fusion and Screening of Positive Hybridoma Cell Lines

The cell fusion and screening of the positive hybridoma cell lines were similar to those previously reported in the literature [43,44], with some modifications. Specifically, the splenocytes were isolated and fused with NS0 myeloma cells at a 10:1 ratio using PEG 1500 as a fusing agent. Ten to fourteen days after cell fusion, both inELISA and icELISA were employed to screen the positive hybridomas obtained from supernatants, and the positive hybridomas after expanded culture were subcloned thrice using the limiting dilution method. The colonies of interest were then frozen in a culture medium containing 10% DMSO, cryopreserved in liquid nitrogen, and labeled.

#### 4.7.3. Production and Purification of Cr(III)-EDTA mAb

The in vivo-induced ascites method was adopted to mass produce Cr(III)-EDTA mAb [45]. Two mature female Balb/c mice were injected intraperitoneally with 1.0 mL of FIA, and twelve days later, injected with 1 to 5 × 10^6^ hybridoma cells. After ten days, the abdomen of the mice was enlarged to collect ascites, which was then purified by the saturated ammonium sulfate precipitation method.

### 4.8. Assessment of Cr(III)-EDTA mAb

#### 4.8.1. Analysis of Isotype and Stability of Hybridoma Cell Lines

The isotypes of the mAbs prepared were identified using a commercially available mouse mAb isotype kit. The cryopreserved hybridoma cells were resuscitated and passaged once every ten days for five times. Both inELISA and icELISA were used to detect the antibody titers and IC50 values of Cr(III)-EDTA in the supernatants at different passages to determine the stability of the hybridoma cells secreting an antibody [46]. 

#### 4.8.2. Identifying the Affinity, Sensitivity and Specificity of Cr(III)-EDTA mAbs

The Batty saturation method was employed to determine the affinity of the ZEN mAbs [47], as follows: Ka = (n − 1)/[2(n[Ab’]t − [Ab]t)].(3)
where n is [Ag]/[Ag’], [Ag]t and [Ag’]t indicate the different concentrations of a coating antigen, and [Ab]t and [Ab’]t represent the corresponding 50% Amax value of the ZEN mAb concentration with different coating antigen concentrations. 

The ZEN mAb with the highest titer, the best inhibition, and the maximum affinity in each group was filtrated for development of icELISA. The sensitivity and specificity of Cr(III)-EDTA mAbs were assessed (For detailed results, see “Section 2.8.1. Sensitivity and Section 2.8.2. Specificity in Section 2.8. Validation of icELISA”). 

### 4.9. Development and Optimization of IcELISA

#### 4.9.1. IcELISA Procedures

The icELISA procedure was similar to that described previously [24,48]. The coating antigen Cr(III)-iEDTA-OVA was diluted in CBS at 2 μg/mL, added to the microplate at 100 μL/well, and incubated at 37 °C for 2 h. The microplate was washed thrice using PBST, and unbound active sites were blocked with 250 µL/well of blocking buffer at 37 °C for 1 h or at 4 °C overnight. The microplate was then washed, and 50 μL/well of Cr(III)-iEDTA mAb at an appropriate dilution in HBS was added, followed by a 50 µL/well addition of serial dilutions of competitors in HBS, and incubated at 37 °C for 30 min. Next, 50 μL/well of GaMIgG-HRP was added and then incubated at 37 °C for 30 min. After six washes, 100 µL/well of freshly prepared TMB solution was added, then incubated at room temperature for 10 min. The reaction was stopped by adding 100 μL/well of 2 M H_2_SO_4_, and the absorbance value was measured at 450 nm. Pre-immunization of serum and PBST were used as negative and blank controls, respectively, and each sample was incubated as duplicates in at least three independent experiments.

#### 4.9.2. IcELISA Optimization

A checkerboard titration procedure was employed to determine the optimal dilutions of Cr(III)-iEDTA-OVA, Cr(III)-EDTA mAb and GaMIgG-HRP to promote the performance of the icELISA. The well with an absorbance value of 1.0 at 450 nm was defined as the optimal working concentrations for icELISA. The ratio calculated via regression equation was used as the evaluation criteria, and the effects of the assay buffer, ionic strength, and pH value on the performances of icELISA were studied. The effect of the ionic strength was evaluated by preparing Cr(III) standard solutions and diluting the Cr(III)-EDTA mAb in HEPES-KCl buffer solution (10 mM HEPES and 3 mM KCl), which had been augmented with varying NaCl concentration (0.137, 0.2, 0.4, 0.8, and 1.6 M NaCl). The effect of the HBS buffer solution with different pH values (5.0, 6.0, 7.4, 8.0 and 9.0) was tested and optimized [49,50].

#### 4.9.3. IcELISA Standard Curve

The Cr(III)-EDTA standard stock solution was diluted with PBS at various concentrations (0.625, 1.25, 2.5, 5.0, 10.0, 20.0, 40.0, 80.0 and 160.0 μg/L), and a nine-point standard curve of icELISA under the above-optimized conditions was achieved by plotting the gradient concentrations (Log C) versus the inhibition percentage values, and the inhibition percentage was calculated using the equation [51,52]: Inhibition percentage (%) = (1 − B/B0) × 100.(4)
where B is the absorbance value of a tested sample solution, and B0 is the absorbance value of a similar solution without Cr(III)-EDTA.

### 4.10. Validation of IcELISA

#### 4.10.1. Sensitivity

The sensitivity of the icELISA was determined by identifying the LOD based on the mean value of 20 blank samples plus 3-fold standard deviation (SD). The working linear range of the icELISA was defined as a 20% to 80% inhibition rate (IC20 to IC80) [53,54].

#### 4.10.2. Specificity

A CR test was carried out to determine the specificity of icELISA for analytes, including Cr(III)-EDTA, EDTA, and other metal ions chelate with EDTA such as Cd(II)-EDTA, Hg(II)-EDTA, Pb(II)-EDTA, Cu(II)-EDTA, Mn(II)-EDTA, Mg(II)-EDTA, Zn(II)-EDTA, Fe(III)-EDTA, and Al(III)-EDTA. The CR was calculated as follows [55,56]: CR (%) = [IC50 (Cr(III) − EDTA)/IC50 (other metal ion chelate)] × 100%.(5)

#### 4.10.3. Accuracy and Precision

The accuracy and precision were assessed by the spike-and-recovery test and the coefficient of variation (CV), respectively. The blank samples (soil, wheat flour, rice, and pig feed) in six replicates were spiked Cr(III) at three concentration levels (20, 40, 80 μg/L) and analyzed by the icELISA. The recovery was calculated using the equation (5). The precision was evaluated via repeated analysis of the spiked samples and by comparison of the intra- and inter-assay coefficient of variations. Intra-assay variation was measured by six replicates of each spiked concentration and the inter-assay variation was based on the results of six different days [57,58].
Recovery (%) = (concentration measured/concentration spiked) × 100.(6)

#### 4.10.4. Reliability

The reliability was performed by comparing the icELISA and ICP-OES analysis methods. One hundred actual samples (20 samples each for soil, rice, wheat flour, and pig feed) were purchased from the local market. Four known positive samples (two wheat samples and two pig feed samples) were provided by the Feed Safety Quality Control Center of the Ministry of Agriculture. All samples were divided into two groups, one group was determined by icELISA, and the other group was determined by ICP-OES, as described by Aghamohammadi et al. [57]. The correlation between icELISA and ICP-OES analysis was calculated.

### 4.11. Preparation of Standard Solution and Sample Solution

#### 4.11.1. Standard Solution

1.0 mL Cr(III) standard solution (1000 μg/L) and an equal volume of EDTA (1 M) were mixed, left to stand for 10 min, and the mixture was diluted with HBS to the required concentration.

#### 4.11.2. Sample Solution

The sample solution was prepared by the wet acid digestion method [29,59]. One gram of each sample was placed in a 100 mL conical flask, and 10 mL HNO_3_, 10 mL HF and 1.0 mL HClO_4_ were added successively. The resulting solution was heated at 195 °C until no white smoke was emitted. After cooling to room temperature, 5 mL deionized water and 5 mL HCl (0.1 M) were added, then transferred into a 50 mL centrifuge tube, and centrifuged at 5000 r/min for 10 min, and the supernatant diluted to 50 mL with deionized water. Next, 1.0 mL of the collected supernatant was adjusted to a pH of 9.0 with 1 M NaOH, and 1.0 mL of reducing agent NaHSO_3_ (1 mM) was then added and reacted for 30 min in the presence of 0.5 mM EDTA to chelate the reduced Cr(III). The reaction solution was adjusted to a pH of 7.4 and the reaction was left to continue for 1 h. The resultant solution was the sample solution to be tested.

### 4.12. Statistical Analysis and Image Processing

All the experiments were conducted in triplicate. Origin Pro 2018 (OriginLab corporation, Northampton, MA, USA) and Excel software (Microsoft Corporation, Redmond, WA, USA) were used for plotting the standard curves and data analysis. ChemDraw 20.0 (PerkinElmer Informatics, Inc., Waltham, MA, USA) was employed to sketch the chemical formulas.

## 5. Conclusions

The study successfully synthesized immunogen Cr(III)-iEDTA-BSA with a molecular binding ratio of 15.48:1 using the phenyl isothiocyanate method. High affinity Cr(III)-EDTA mAbs were prepared by Balb/c mouse immunization and hybridoma technology, in which the Ka of Cr(III)-EDTA mAb 2A3 with the highest affinity was 2.69 × 10^9^ L/moL. Based on mAb 2A3, an icELISA for Cr(III)-EDTA detection was established. Under optimal experiment conditions, the LOD of icELISA was 1.0 μg/L, and the detection range was 1.13 to 66.30 μg/L, indicating that the assay had good sensitivity. There is no CR with other compounds except for a slight CR with the Fe(III)-EDTA, exhibiting that the assay had good specificity. The results from the spiking experiment showed that the average intra- and inter-assay recovery rates were 90% to 109.5% and 90.4% to 97.2%, respectively, while the CVs were 11.5% to 12.6% and 11.1% to 12.7%, respectively. This demonstrated that the assay had good accuracy and precision. The preliminary application of the icELISA and comparison with ICP-OES showed that the coincidence rate of the two methods was 100%, and the correlation coefficient was 0.987, and therefore, the assay had good reliability. The Cr(III)-EDTA icELISA established in this study has high sensitivity, strong specificity, and good accuracy, and can be effectively used to detect total chromium residues in food, feed, and environmental samples.

## Figures and Tables

**Figure 1 molecules-27-01585-f001:**
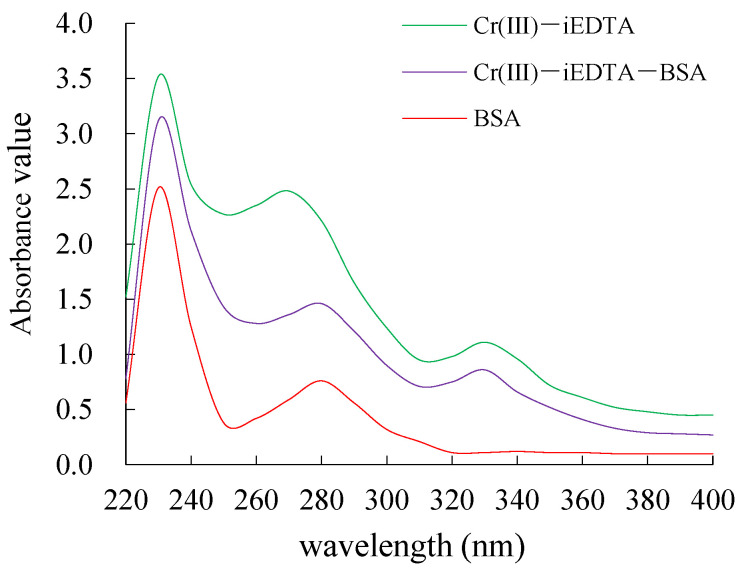
UV spectra of Cr(III)-iEDTA-BSA synthesized using the isothiocyanate method. UV: ultraviolet-visible spectrophotometer. Cr(III)-iEDTA-BSA: Cr(III) is chromium trivalent ion, iEDTA is coupling agent isothiocyanatebenzyl-EDTA, BSA is carrier protein bovine serum albumin, and Cr(III)-iEDTA-BSA is the synthetic immunogen.

**Figure 2 molecules-27-01585-f002:**
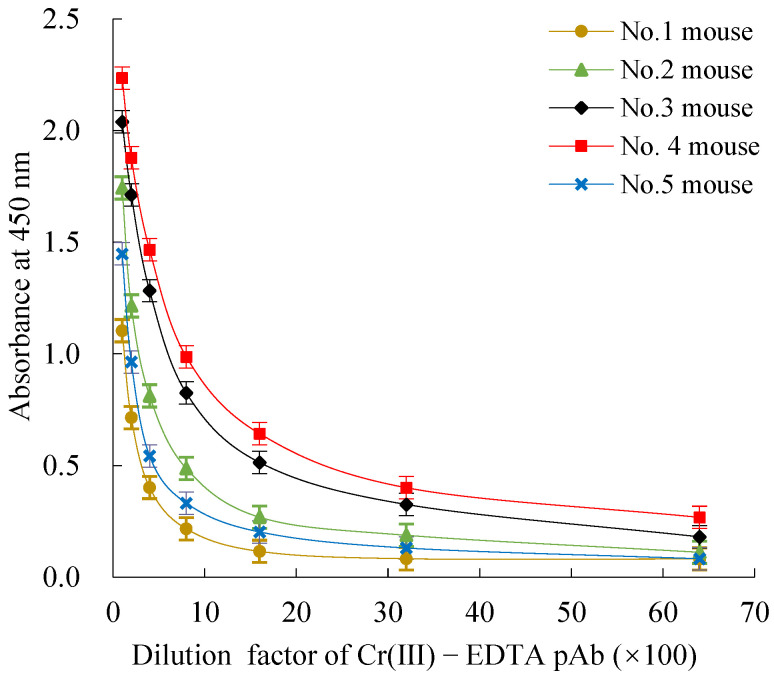
The inELISA titers measurement for Cr(III)-EDTA pAb. InELISA: indirect non-competitive ELISA. Cr(III)-EDTA: Cr(III) is chromium trivalent ion, EDTA is chelating agent ethylenediamine tetraacetic acid, and Cr(III)-EDTA is a chelate of Cr(III) and EDTA. pAb: polyclonal antibody.

**Figure 3 molecules-27-01585-f003:**
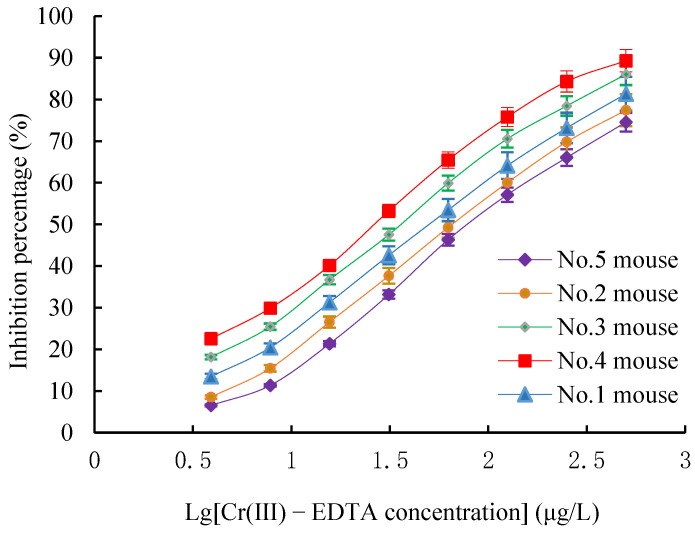
Sensitivity measurement of Cr(III)-EDTA pAb against Cr(III)-EDTA via icELISA. icELISA: indirect competitive ELISA.

**Figure 4 molecules-27-01585-f004:**
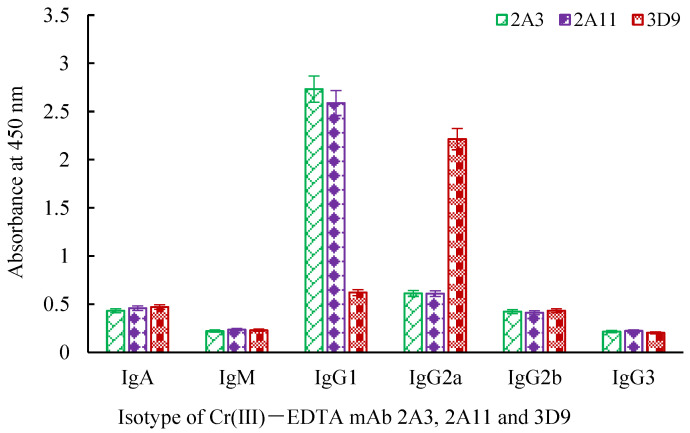
The isotype analysis of Cr(III)-EDTA mAb 2A3, 2A11 and 3D9. mAb: monoclonal antibody.

**Figure 5 molecules-27-01585-f005:**
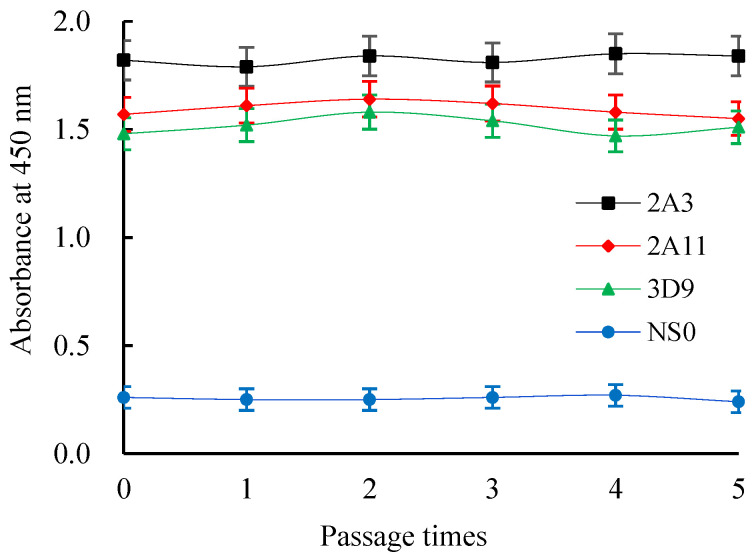
The titers of Cr(III)-EDTA mAbs secreted by three hybridomas after five freeze and thaw cycles.

**Figure 6 molecules-27-01585-f006:**
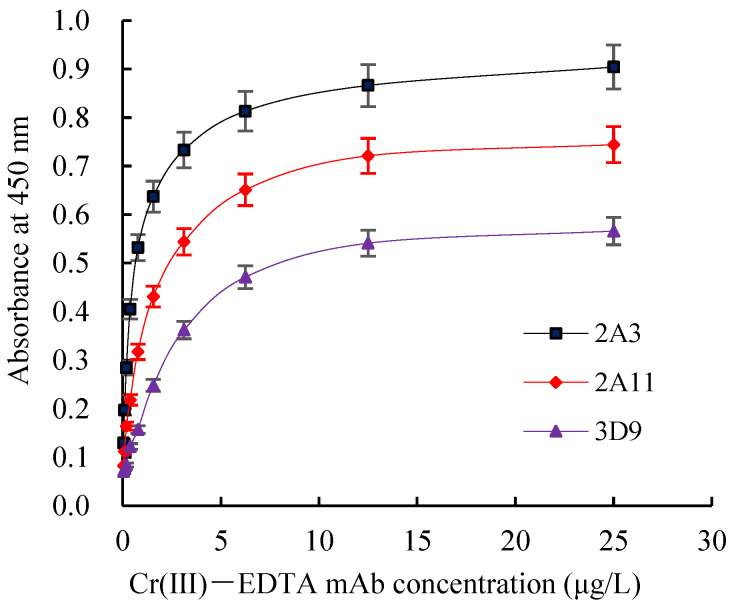
The Ka curves of Cr(III)-EDTA mAbs. Ka: affinity constant.

**Figure 7 molecules-27-01585-f007:**
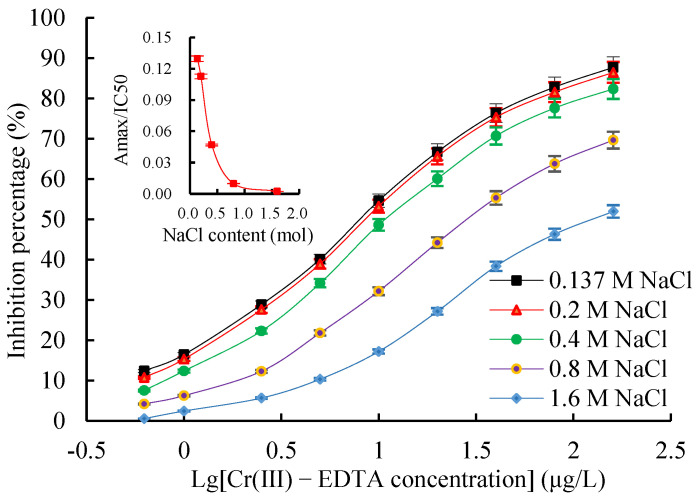
The effects of ionic strength on an icELISA. Insets indicate the fluctuations of Amax/IC50 as a function of ionic strength.

**Figure 8 molecules-27-01585-f008:**
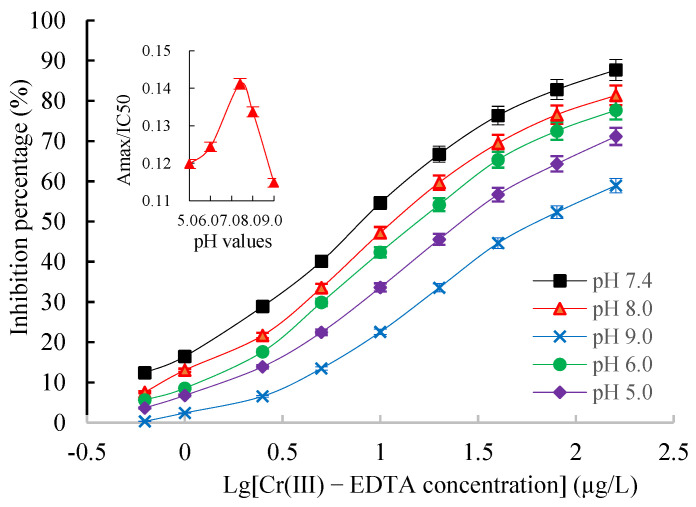
The effects of pH value on an icELISA. Insets indicate the fluctuations of Amax/IC50 as a function of pH value.

**Figure 9 molecules-27-01585-f009:**
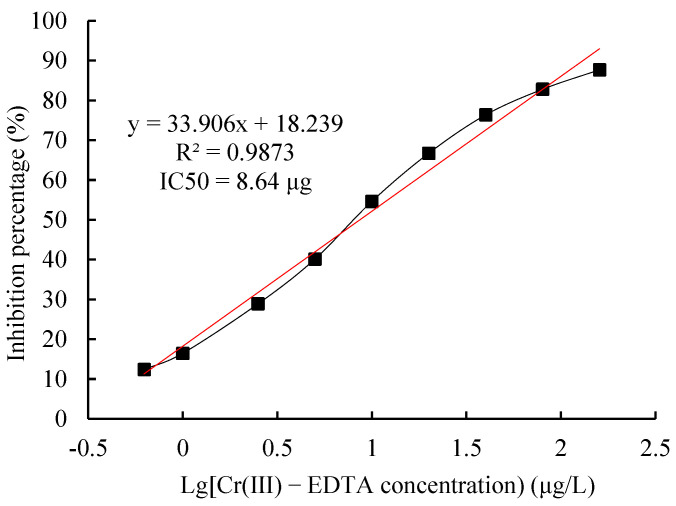
The standard curve of an icELISA for Cr(III)-EDTA. The standard curve was obtained using icELISA optimized conditions for Cr(III)-EDTA.

**Figure 10 molecules-27-01585-f010:**
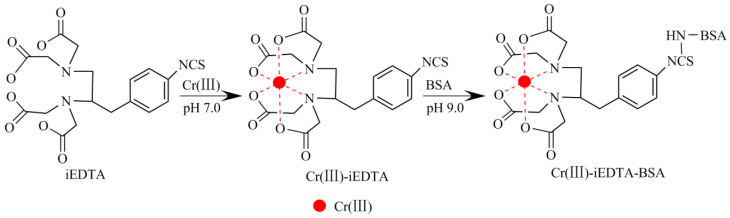
Synthesis scheme of the Cr(III)-iEDTA-BSA via the isothiocyanate method.

**Table 1 molecules-27-01585-t001:** The titers and IC50 values of Cr(III)-EDTA mAbs generated by three hybridomas.

Cr(III)-EDTA mAbs	Titers of Supernatants	Titers of Ascites	IC50 Values of Supernatants (μg/mL) ^a^	IC50 Values of Ascites (μg/mL)
2A3	5.12 × 10^2^	5.12 × 10^5^	9.26	9.04
2A11	2.56 × 10^2^	2.56 × 10^5^	16.56	16.18
3D9	2.56 × 10^2^	1.28 × 10^5^	21.82	20.35

Note. ^a^ IC50 means 50% inhibition concentration.

**Table 2 molecules-27-01585-t002:** IC50 and Amax/IC50 values of Cr(III)-EDTA and 2B6 mAb in different assay buffers.

Cr(III)-EDTA in Buffer	mAb in Buffer	Amax	IC50	Amax/IC50
PBS	PBS	1.82	9.34	0.1949
HBS	HBS	1.74	8.64	0.2014
PBS	HBS	1.78	8.95	0.1989
HBS	PBS	1.79	9.12	0.1963

**Table 3 molecules-27-01585-t003:** The percent cross-reactivity of the icELISA with EDTA and other metal ions.

Metal-Chelates	IC50 (μg/L) ^a,b^	CR (%) ^c,d^
Cr(III)-EDTA	8.64	100
Fe(III)-EDTA	771.43	1.12
Al(III)-EDTA	>3.2 × 10^3^	<0.5
Cd(II)-EDTA	>6.4 × 10^3^	<0.5
Hg(II)-EDTA	>6.4 × 10^3^	<0.5
Pb(II)-EDTA	>6.4 × 10^3^	<0.5
Cu(II)-EDTA	>6.4 × 10^3^	<0.5
Mn(II)-EDTA	>6.4 × 10^3^	<0.5
Mg(II)-EDTA	>6.4 × 10^3^	<0.5
Zn(II)-EDTA	>6.4 × 10^3^	<0.5
EDTA	>6.4 × 10^3^	<0.5

Note. ^a^ IC50 referred to the metal ion concentration that produces a 50% inhibition of the signal. ^b^ All of the data were calculated from triplicate assays, and the average coefficient of variation (CV) was below 10%. ^c^ CR values were calculated as CR(%) = IC50 (Cr(III)-EDTA)/IC50 (other metal-chelates) × 100. ^d^ All of the data were calculated using the CR of Cr(III)-EDTA mAb against Cr(III)-EDTA as 100%.

**Table 4 molecules-27-01585-t004:** Accuracy and precision measurements of the icELISA for Cr(III)-EDTA.

Samples	Cr(III)-EDTA Concerntration (μg/L)	Inner Batch			Among Batches		
		Measured Value (μg/L) ^a^	Recovery (%)	CV (%) ^a,b^	Measured Value (μg/L) ^a^	Recovery (%)	CV (%) ^b^
Soil	2	2.19 ± 0.37	109.5	12.7	2.16 ± 0.33	10.8	13.7
	10	9.12 ± 1.65	91.2	11.3	9.04 ± 1.72	90.4	12.4
	50	41.62 ± 3.28	83.2	13.5	42.31 ± 3.41	84.6	11.8
Wheat flour	2	2.08 ± 0.27	104	12.9	2.11 ± 0.24	105.5	13.3
	10	9.33 ± 1.48	93.3	11.8	9.11 ± 0.1.51	91.1	11.1
	50	43.61 ± 3.13	87.2	9.7	44.53 ± 2.51	89.1	9.6
Rice	2	2.05 ± 0.31	102.5	13.3	2.12 ± 0.15	106	12.4
	10	9.24 ± 1.56	92.4	12.4	8.91 ± 0.18	89.1	10.7
	50	44.25 ± 0.38	88.5	9.1	48.22 ± 2.54	96.4	10.3
Pig feed	2	1.97 ± 0.32	98.5	13.8	1.95 ± 0.27	97.5	13.6
	10	8.73 ± 1.74	87.3	12.6	8.82 ± 1.81	88.2	13.1
	50	42.12 ± 3.28	84.2	11.3	42.77 ± 3.36	85.5	11.4

Note. ^a^ Measured values present the mean of six replicates determination. ^b^ CV means the coefficient of variation.

**Table 5 molecules-27-01585-t005:** Comparison of icELISA with ICP-OES for the determination of Cr(III) in actual samples.

Samples	Sample Number	Positive Sample Number		Positive Rate (%)	Positive Sample Content Range (μg/L)		CV (%)
		IcELISA	ICP-OES		IcELISA	ICP-OES	
Soil	20	4	4	20	1.2~44.2	1.1~42.3	10.6
Rice	20	3	3	15	1.5~23.3	1.3~21.4	11.7
Wheat flour	20	2	2	10	1.4~18.1	1.3~16.7	11.4
Pig feed	20	4	4	20	1.4~36.5	1.2~34.8	13.3
Total	80	13	13	16.3	1.2~44.2	1.1~42.3	10.6~13.3

## Data Availability

Not applicable.

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
