# Peer review of "Indirect Competitive ELISA for the Determination of Total Chromium Content in Food, Feed and Environmental Samples"

_molecules, 2022, doi:10.3390/molecules27051585_

Round 1

Reviewer 1 Report

The submitted manuscript (Manuscript Number: molecules-1569179) entitled “An indirect competitive ELISA for determination of total chromium content in food, feed and environmental samples” is written in a confusing and disordered manner. Its the referee opinion that the manuscript should be considered under “major revision” before publication. The suggestions that are needed to be taken into consideration are as follows:

Comments:

  1. This type of similar work has already been published elsewhere (Zou, J., Tang, Y., Zhai, Y., Zhong, H., & Song, J. (2013).  Analytical Methods, 5(11), 2720). So, referee could not find any novelty in this work. Specify the difference between previously published work (as above cited) with present work.
  2. Abbreviations must be explained where it is first used in the manuscript.
  3. Several discrepancies occur in writing chemical formulaes such as, H2SO4, CrCl3.6H2 Authors must revise them in the manuscript.
  4. Authors must rewrite all the measurements values in same significant figures. (1 mL or 1.0 mL)
  5. The caption of Figure 1 must be revised and written carefully.
  6. Equation and denotions of terms in equation A= eCL (Line 152, Page 4) must be corrected and molar extinction coefficient (e) must be incorporated.
  7. All the equations must be properly numbered.
  8. In Section 2.6.2, specify the dilution factor with HBS.
  9. Mention the Section no. along with Section name in Line 209 Page 5.
  10. The values of titres in Section 3.3 must be rechecked.
  11. The repetition of line that “all the experiments are conducted in triplicate” should be avoided in each figure caption and mentioned at a single place in materials and methods section.
  12. The authors must check the manuscript for grammatical errors. (Many statements are written without proper endings. (Line 119, Page 3; Line 425, Page 13), several vocabulary mistakes are there in the manuscript which must be resolved.)

Author Response

Responses to Reviewer 1

  1. This type of similar work has already been published elsewhere (Zou, J., Tang, Y., Zhai, Y., Zhong, H., & Song, J. (2013).  Analytical Methods, 5(11), 2720). So, referee could not find any novelty in this work. Specify the difference between previously published work (as above cited) with present work.

Thanks for your valuable advice . According to your suggestion, we have discussed  this issue and marked them in yellow on lines 147-150, hoping that you can accept our  point of view.

  1. Abbreviations must be explained where it is first used in the manuscript.

Thank you for your valuable guidance . According to your guidance, we have revised these errors and marked them in yellow on lines 11, 12, 16, 17, and 26.

  1. Several discrepancies occur in writing chemical formulaes such as, H2SO4, CrCl3·6H2O. Authors must revise them in the manuscript.

We are sorry for our careless, and we have corrected them carefully and marked them in yellow on lines 419 and 560.

  1. Authors must rewrite all the measurements values in same significant figures. (1 mL or 1.0 mL)

We are sorry for our careless, and we have corrected them carefully and marked them in yellow on lines 254, 529, 559, 560, 563, 565, 573, 574, 642, 794, 798, 799, and 831.

  1. The caption of Figure 1 must be revised and written carefully.

Thanks for your valuable advice, according to your suggestion, we have modified the title in Figure 10, as indicated in yellow on line 611.

  1. Equation and denotions of terms in equation A= eCL (Line 152, Page 4) must be corrected and molar extinction coefficient (e) must be incorporated.

We are very sorry, the submission system does not recognize the letter ε due to our wrong operation method, we have corrected it in yellow on line 618.

  1. All the equations must be properly numbered.

Thanks for your valuable advice, and according to your suggestion, we have numbered the six equations in the full text in order, and marked them in yellow on lines 618, 654, 740, 799, 812, and 821.

  1. In Section 2.6.2, specify the dilution factor with HBS.

We are very sorry for us not explaining clearly. In fact, we have carried out a buffer screening experiment, and the result is that HBS is better than PBS. The test results are listed in Table 2 in line 313 marked in yellow.

  1. Mention the Section no. along with Section name in Line 209 Page 5.

Thank you for your valuable guidance . According to your guidance, we have revised and marked them in yellow on lines 746-747.

  1. The values of titres in Section 3.3 must be rechecked.

We are very sorry for our careless. The experimental result is that the titer of mouse 4 is the highest and the inhibitory effect is the best, and we have corrected it and marked errors in yellow in the text on lines 204, 206, and 207.

  1. The repetition of line that “all the experiments are conducted in triplicate” should be avoided in each figure caption and mentioned at a single place in materials and methods section.

Thanks for your valuable advice. According to your suggestion, we put the sentence "All the experiments were conducted in triplicate in section “2.12. Statistical analysis and image processing.” on line 845. At the same time, all the contents involved in the figures have been deleted and modifications are marked in yellow on lines 209, 211, 256, 266, 272, 315, 318, and 337.

  1. The authors must check the manuscript for grammatical errors. (Many statements are written without proper endings. (Line 119, Page 3; Line 425, Page 13), several vocabulary mistakes are there in the manuscript which must be resolved.)

We are very sorry for our careless. We have carefully revised and marked in yellow on lines 415, 421, 448, 459, 578, and the like.

Reviewer 2 Report

Comments to Authors

The present paper, “An indirect competitive ELISA for determination of total chromium content in food, feed, and environmental sample,” tests total chromium content in food, feed, and environmental samples. It seems to use and develop a novel indirect competitive ELISA method to apply in food, feed, and environmental samples. Some crucial comments should be concerned as follows.

  1. The present study should clearly state the rationale of this indirect competitive ELISA method in the Introduction section.
  2. The novelty and characteristics of the present indirect competitive ELISA method should be explained and described in the Introduction and/or Discussion sections.
  3. What is the function of the present indirect competitive ELISA method? It needs to clarify.
  4. Compare the different and the same parts between/among the previous ELISA and the present indirect competitive ELISA, and then state these points in the Discussion section.
  5. What are the advantages and disadvantages of the present indirect competitive ELISA? This point should be clarified in the Discussion section.
  6. What are the limitations of the present indirect competitive ELISA? It is required to clarify.

The manuscript should be major revised. The current status of the manuscript cannot be considered for acceptance.

Author Response

Responses to Reviewer 2

  1. The present study should clearly state the rationale of this indirect competitive ELISA method in the Introduction section.

Thanks for your valuable advice. According to your suggestion, we carefully study and summarize the principle of icELISA, which is explained in the Introduction and marked in gray in lines 86-137.

  1. The novelty and characteristics of the present indirect competitive ELISA method should be explained and described in the Introduction and/or Discussion sections.

Thanks for your valuable advice. According to your suggestion, In the Intruduction, we supplemented the advantages and disadvantages of icELISA compared with other physicochemical analysis methods and immunoassay methods, and its application in the detection of heavy metal ions. The modifications are marked in gray on lines 137-146. We hope you can accept our supplement.

  1. What is the function of the present indirect competitive ELISA method? It needs to clarify.

Thanks for your valuable advice. According to your suggestion, In the Disscussion section, we have supplemented the introduction of function of icELISA, and marked them in gray on lines 485-500.

4.Compare the different and the same parts between/among the previous ELISA and the present indirect competitive ELISA, and then state these points in the Discussion section.

Thank you very much. According to your guidance, In the discussion section, we added the different parts and the same parts of dcELISA and icELISA. The revised parts are marked in gray on lines 501-525.

  1. What are the advantages and disadvantages of the present indirect competitive ELISA? This point should be clarified in the Discussion section.

Thanks for your valuable advice. According to your suggestion, In the Disscussion section, we have supplemented the advantages and disadvantages of dcELISA and icELISA, and revised parts are marked in gray on lines 528-532.

  1. What are the limitations of the present indirect competitive ELISA? It is required to clarify.

Thanks for your valuable advice. According to your suggestion, we have supplemented the limitations of the present icELISA, and revised parts are marked in gray on lines 532-534.

Reviewer 3 Report

The manuscript of Wang et al, entitled „An indirect competitive ELISA for determination of total chromium content in food, feed and environmental samples”, aims at determining chromium content in different samples. It describes extensively the development of a monoconal antibody with EDTA-chelated Cr(III) conjugated to BSA, and using this antibody the development of a competitive ELISA method based on EDTA-chelated Cr(III) conjugated to ovalbumin as target antigen. The synthesis and characterization of the antigen/immunogen, the method of immunizing and the development and complete immunological characterization of the hybridoma and of the monoclonal antibody are all adequately described. Indirect competitive ELISA has been developed, optimized and validated. The Cr(III) content of samples measured by competitive ELISA and optical electron spectroscopy has been shown to correlate.

The research described is highly valuable, the data are mostly well presented. On the other hand, quite similar Cr(III) detecting competitive ELISA methods have been described before, although with different ELISA detection methods, such as the work of Yao et al (cited as Ref 17) and a publication of the same group, Yu et al, A portable chromium ion detection system based on a smartphone readout device, Anal. Methods, 2016,8, 6877-6882. It would be interesting to read Authors’ comparison as to the advantages of their methods, especially as Authors emphasise to usefulness of easily usable ELISA kit. 

Reviewer does not agree with calling the y axis of icELISA results „Inhibition percentage” (Fig4, Figs 8-9-10), 100 % inhibition is, when binding is zero, as when the Cr(III)-EDTA concentration is small. In my opinion it is the binding percentage of the antibody on the axis, as shown by labelling the B/B0, and the diminishing binding indicates the inhibition. I would suggest that Inhibition % = (1-B/B0)x100% should be calculated and applied on the graphs.

Line 44-46: „Under normal physiological conditions, Cr(VI) enters cells after ingestion and can be reduced to Cr(V), Cr(IV), Cr(III), sulfur group, hydroxyl radical, etc.,” there must be some problem with this sentence, where do sulfur group and hydroxyl radical come from?

Line 388: I suppose that instead of Cr(III) authors wanted to write EDTA (or maybe some chelator).

Figure 3: According to Figure 3 Mouse 4 seems to have the highest titer, not Mouse 3, as stated in the text. It may be practical to have the same color code for the mice in Figures 3 and 4.

Lines 168-169: It is not absolutely clear whether the immunization was performed at weeks 0, 4, 8, 12, 16 (to me 4 weeks interval would mean that), or five times during 4 weeks, e.g. weeks 0, 1, 2, 3, 4.

Table 2: Throughout the manuscript Cr(III) is used, only in Table 2 it is Cr3+.

Line 130: Water polishing system is not the right word for a water purification (filtering?) equipment.

After addressing the above comments and correcting these minor problems the manuscript of Wang et al may be suitable for publication in Molecules. 

Author Response

Responses to Reviewer 3

  1. Reviewer does not agree with calling the y axis of icELISA results „Inhibition percentage” (Fig4, Figs 8-9-10), 100 % inhibition is, when binding is zero, as when the Cr(III)-EDTA concentration is small. In my opinion it is the binding percentage of the antibody on the axis, as shown by labelling the B/B0, and the diminishing binding indicates the inhibition. I would suggest that Inhibition % = (1-B/B0)x100% should be calculated and applied on the graphs.

Thanks for your valuable advice. According to your suggestion, we have modified the y-axis B/B0% of the inhibition rate calculation method in Figures 3, 7, 8, and 9 to be (1-B/B0) × 100%, which can better reflect the true meaning of the inhibition rate. Modifications are marked in green on lines 211, 315, 318, and 337.

  1. Line 44-46: „Under normal physiological conditions, Cr(VI) enters cells after ingestion and can be reduced to Cr(V), Cr(IV), Cr(III), sulfur group, hydroxyl radical, etc.,” there must be some problem with this sentence, where do sulfur group and hydroxyl radical come from?

We are sorry for our wrong explanation. According to your suggestion, and we have corrected this sentence and and marked it in green on lines 62-63.

  1. Line 388: I suppose that instead of Cr(III) authors wanted to write EDTA (or maybe some chelator).

We are very sorry for our careless. Thank you for your seriousness and care, we have revised to chelating agent EDTA, marked in green on line 409.

  1. Figure 3: According to Figure 3 Mouse 4 seems to have the highest titer, not Mouse 3, as stated in the text. It may be practical to have the same color code for the mice in Figures 3 and 4.

We are very sorry for our careless. The experimental result is that the titer of mouse 4 is the highest and the inhibitory effect is the best, and we have corrected it and marked errors in yellow in the text on lines 204, 206, and 207.

  1. Lines 168-169: It is not absolutely clear whether the immunization was performed at weeks 0, 4, 8, 12, 16 (to me 4.4 weeks interval would mean that), or five times during 4 weeks, e.g. weeks 0, 1, 2, 3, 4.

We are very sorry for our unclear express. We have split this long sentence into three short sentences to make it clearer. Modified sections are marked in green on lines 662-666.

  1. Table 2: Throughout the manuscript Cr(III) is used, only in Table 2 it is Cr3+.

Thank you for your careful reminder. We have revised Cr3+ to Cr(III), and marked it in Table 3 on line 352.

  1. Line 130: Water polishing system is not the right word for a water purification (filtering?) equipment.

Thanks for your valuable advice, and according to your suggestion, we have deleted the word “polishing” and marked this sentence in green on line 595.

  1. After addressing the above comments and correcting these minor problems the manuscript of Wang et al may be suitable for publication in Molecules. 

Thank you for your affirmation and encouragement of the manuscript, and  Thank you again for your valuable suggestions.

Round 2

Reviewer 1 Report

Authors clearly answers all raised questions so now this manuscript may be published in this journal.

Reviewer 2 Report

Dear Authors,

Greetings for the day.

I found that all comments have been fully revised.

Thank you for your submission.

Thank you.

Best regards,

Andrew